# Combined abnormal muscle activity and pain-related factors affect disability in patients with chronic low back pain: An association rule analysis

Hayato Shigetoh[1,2]*, Yuki Nishi[1], Michihiro Osumi[3], Shu Morioka[1,3]

1 Department of Neurorehabilitation, Graduate School of Health Sciences, Kio University, Koryo-cho, Nara, Japan, 2 Miura Internal Medicine Michiko Pediatrics Clinic, Marugame-shi, Kagawa, Japan, 3 Neuro Rehabilitation Research Center, Kio University, Koryo-cho, Nara, Japan

* hayato.pt1121@gmail.com

## Abstract

### Objectives

In patients with chronic low back pain (CLBP), reduced lumbar flexion-relaxation and reduced variability of muscle activity distribution are reported as abnormal muscle activity. It is not known how abnormal muscle activity and pain-related factors are related to CLBP-based disability. Here, we performed an association rule analysis to investigated how CLBP disability, muscle activity, and pain-related factors in CLBP patients are related.

### Methods

Surface electromyographic signals were recorded from over the bilateral lumbar erector spinae muscle with four-channel electrodes from 24 CLBP patients while they performed a trunk flexion re-extension task. We calculated the average value of muscle activities of all channels and then calculated the flexion relaxation ratio (FRR) and the spatial variability of muscle activities. We also assessed the pain-related factors and CLBP disability by a questionnaire method. A clustering association rules analysis was performed to determine the relationships among pain-related factors, the FRR, and the variability of muscle activity distribution.

### Results

The association rules of severe CLBP disability were divisible into five classes, including 'low FRR-related rules.' The rules of the mild CLBP disability were divisible into four classes, including 'high FRR-related rules' and 'high muscle variability-related rules.' When we combined pain-related factors with the FRR and muscle variability, the relationship between abnormal FRR/muscle variability and CLBP disability became stronger.

**Data Availability Statement:** All relevant data are within the manuscript and its Supporting information files. Also, the data are held in Figshare (https://doi.org/10.6084/m9.figshare.13246925).

**Funding:** The author(s) received no specific funding for this work.

## Discussion

Our findings thus highlight the importance of focusing on not only the patients' pain-related factors but also the abnormal motor control associated with CLBP, which causes CLBP disability.

## Introduction

Low back pain (LBP) is one of the common symptoms and the most common cause of disability worldwide [1–3]. The severity and disability of LBP are associated with several factors including psychological, social, and biophysical factors, co-morbidities, and pain-processing mechanisms [4]. Interestingly, psychological factors such as anxiety, depression, catastrophic thinking, kinesiophobia, and self-efficacy are often associated with disability in individuals with chronic low back pain (CLBP) [5–11]. In recent years, some research groups have reported that body perception disturbance [12] (i.e., lumbar proprioceptive deficits, body image disorder, and neglect-like symptoms) and central sensitivity syndromes (i.e., symptoms associated with central sensitization despite no structural pathology) [13] also affect disability in CLBP patients. It is thus important to examine multiple factors when seeking to improve the disability of individuals suffering from CLBP.

The abnormal muscle activities in CLBP patients should also be considered. Studies of CLBP patients reported that the patients' lumbar flexion-relaxation (FR) and variability of muscle activity distribution, which were observed by electromyography (EMG), were distorted [14–16]. Flexion-relaxation was described as a reduction in the electrical activity of the superficial spinal extensor muscles in the end range of lumbo-pelvic flexion [17]. An individual's FR value can be objectively calculated as the flexion-relaxation ratio (FRR). In CLBP patients, the FRR is reportedly decreased, indicating that they cannot relax their spinal extensor muscles at the end range of lumbo-pelvic flexion [15, 17].

The variability of muscle activity distribution in CLBP patients is assessed by multi-channel EMG because the limited information that can be obtained from a single pair of electrodes placed over a small muscle region [18]. It has been reported that compared to non-LBP patients, the variability of muscle activity distribution is reduced [19, 20]. These studies also indicated that the regions of muscle activation in CLBP patients were restricted and cranial regions were activated than caudal regions. This decreased motor variability is associated with tight motor control [21]. Increased trunk muscle activation to tighten control comes at the cost of increased spinal loading [22]. Numerous investigations have demonstrated an important relationship between the FRR and both physical factors and pain-related factors in CLBP, including the lumbar flexion angle [23], fatigue [24], cognitive-emotional factors (i.e., fear [23] and cognitive loading [25]), and disability [26–28]. Several research groups have also described relationships between the variability of muscle activity distribution and physical factors in CLBP, including fatigue [19, 20, 29]. Focusing on the pain-related factors, one research group has also described relationships between the variability of muscle activity distribution and pain-related factors in CLBP, including pain severity [30] and disability [30]. However, the relationships between the FRR and the variability of muscle activity distribution and pain-related factors have not been established within the same study, and these relationships remain controversial [26, 30, 31].

Motor control dysfunction and pain-related factors may have different impacts on CLBP disability, and these impacts may differ among subgroups of patients with CLBP. The

identification of such subgroups may inform tailored management, but it has been unclear how abnormal muscle activity and pain-related factors are related to CLBP disability. We hypothesized that the FRR and muscle variability are simply associated with CLBP disability, and we speculated that the relationships among them would become much stronger when pain-related factors are considered in the analyses. We conducted the present study to test our hypothesis by conducting an association rules analysis; this type of analysis reveals causal relationships based on probabilities.

## Patients and methods

### Patients

Twenty-four patients with CLBP (10 men, 14 women) aged 49–85 years (71.6 ± 9.2 yrs, mean ± standard deviation [SD], men: 71.5 ± 6.6, women: 71.7 ± 10.7) were referred from physiotherapy practices and enrolled if they were suffering from LBP lasting >3 months [32]. Patients whose LBP area was defined as the area bounded by the lowest palpable ribs superiorly and the gluteal folds inferiorly were included [33]. The patient's LBP area's detail was a low back area (n = 12) and between sacroiliac joint and gluteal folds inferiorly (n = 12). The patients had referred leg pain (n = 2) and no referred leg pain (n = 22). Also, the patient's LBP was categorized as non-specific LBP (n = 6) and specific LBP (n = 18). The specific LBP included spinal stenosis (n = 3) and lumbar osteoarthritis (n = 15). On the other hand, Patients were excluded if they had any central nervous system disease, dementia, LBP that had appeared within 3 months, difficulty understanding questionnaires and tasks, or difficulty performing tasks. Patients who reported that their LBP condition suddenly worsened were also excluded. The study protocol conformed to the Declaration of Helsinki. The participants provided written informed consent before the study began. This study was approved by the ethics committee of Kio University Health Sciences Graduate School (approval no. H30-06). This study was conducted between July 2019 and December 2019.

### Evaluations of the patients' characteristics by questionnaires

The following characteristics were assessed for each patient: demographic data (age, gender, pain duration), pain intensity (by a numerical rating scale (NRS) for pain, and the Short-Form McGill Pain Questionnaire-2 [SFMPQ-2]), body perception disturbance (by the Fremantle Back Awareness Questionnaire [FreBAQ]), central sensitivity syndromes (by the Central Sensitization Inventory-9 [CSI-9]), psychological conditions (an NRS for fear, plus the Pain Self Efficacy Questionnaire-2 [PSEQ-2], the Hospital Anxiety and Depression Scale [HADS], the Pain Catastrophizing Scale-4 [PCS-4], and the Tampa Scale for Kinesiophobia-11 [TSK-11]), and disability (by the Rolland Morris Disability Questionnaire [RMDQ]).

The NRS for pain was used to assess pain intensity (0: no pain, 10: worst pain imaginable). The SFMPQ-2 was used to assess pain intensity and includes items that assess 22 qualities of pain and the intensity of each quality on an 11-point numerical rating scale [34]. The total score is calculated from the sum of the 22 items. A higher score indicates more severe pain. We used the CSI-9 to assess health-related symptoms that are common to central sensitivity syndromes [35]. The CSI-9 is a shorter version of a 25-item CSI and contains nine items. Higher scores indicate more severe central sensitivity syndrome.

The FreBAQ was applies to assess the patients' the body perception disturbance. FreBAQ is comprised of nine items and three subscales (proprioception, body image, and neglect-like symptoms) [36]. A higher score indicates more severe body perception disturbance. The NRS for fear was administered to assess the patients' fear about lumbar movement (0: no fear, 10: worst fear imaginable). Higher scores indicate more severe lumbar movement fear. The

PSEQ-2 was used to assess self-efficacy about pain. The PSEQ-2 is a shorter version of a 10-item PSEQ and contains two items [37]. A lower score indicates low self-efficacy. We used the HADS to assess anxiety and depression as psychological factors. The HADS contains 14 items and two subscales. The two subscales independently assess depression and anxiety [38]. Higher scores indicate more severe anxiety and depression.

The PCS-4 was used to assess catastrophic thinking; it is a shorter version of a 13-item PCS and contains four items [39]. Higher scores indicate more severe catastrophic thinking. We used the TSK-11 to assess kinesiophobia as a psychological factor. The TSK-11 is a shorter version of a 17-item TSK and contains 11 items [40]. Higher scores indicate more severe kinesiophobia. The RMDQ was used to assess LBP-related disability. It contains 24 items [41], and higher scores indicate more severe disability.

## Standing trunk flexion and re-extension task

Each of the 24 patients was asked to perform a standing trunk flexion and re-extension task while EMG was recorded. The phases of this task were classified into the standing phase, flexion phase, full flexion phase, and extension phase, with each phase lasting 3 sec (Fig 1) [42].

The patient began the task by standing without movement (the standing phase) with his or her feet at hip width. After the first auditory signal, the patient bent forward with a slow and controlled movement (flexion phase) to reach maximal trunk flexion before a second auditory signal. The patient was then asked to maintain the full flexion position (full flexion phase) until a third auditory signal. After the third auditory signal, the patient returned to the upright posture for 3 sec. The interval between each auditory signal was 3 sec. After completing a reference trial at least once, each patient repeated the task for three trials.

## Trunk kinematic recording and analysis

Trunk kinematics were recorded using an accelerometer (AX-3, Axivity, Newcastle upon Tyne, UK). This accelerometer has been validated in a previous study [43]. An accelerometer was fixed over the patient's twelfth thoracic (T12) spinous process with adhesive tape (Fig 2). The sampling frequency was set at 100 Hz. The trunk range of motion was measured as the angular inclination of the trunk at T12 [14]. In this study, the angle of trunk flexion was calculated using the angle between the gravity vector and the z-axis vector (in this study, the direction along the trunk) [44]. The angle values correspond to the range of motion between the patient's starting position before the standing trunk flexion re-extension task and the maximal trunk flexion after the onset of the trunk flexion task. These data were analyzed with custom-

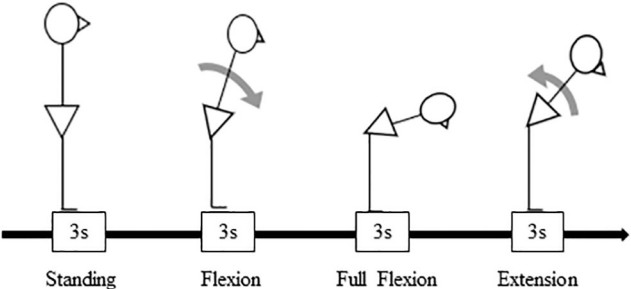

**Fig 1. The standing trunk flexion and re-extension task.**

**Fig 2. The approximate positioning of the EMG grid and accelerometer.** The EMG electrode grid was placed 3 cm lateral to the lumbar spinous process on the bilateral erector spinae. The accelerometer was placed at the 12th thoracic spinous process.

written MATLAB code (v.2019b, MathWorks, Natick, MA, USA). The maximum flexion angle was calculated as the mean maximum flexion angle of the three trials.

## Recording and analysis of muscle activities

Surface EMG signals were detected with a grid of sheet-type electrodes (Unique Medical Co., Brooklyn, NY). The grid consisted of five electrodes (four-channel, with a 25-mm inter-electrode distance in both directions). The patient's glabrous skin over the erector spinae was abraded and cleaned with an alcohol wipe. Two grids of sheet-type electrodes coated with an electroconductive gel were taped to the skin at that site; the electrode was located 3 cm outside the midpoint of the lumbar spinous process [45] and covered upward along the erector spinae from the level of the Jacoby line (Fig 2). A reference electrode was placed over the left-side radial styloid process. The recorded EMG signals (sampling rate: 1000 Hz) were analyzed after band pass filtering (10 and 400 Hz). The root mean square (RMS) EMG values were calculated at each phase of movement in both the reference trial and experimental trials. Then, normalized RMS EMG values in each phase were respectively obtained with the division of the mean RMS value of the experimental trials by the mean RMS value obtained in the extension phase of the reference trial [46]. If the noise could not be removed, the RMS value was excluded from the analysis.

For the calculation of the FRR, the maximum surface EMG value in the flexion phase and the mean surface EMG value in the full flexion phase were calculated as described [15]. The FRR was calculated for each channel as follows: the maximum surface EMG value in the flexion phase was divided by the mean surface EMG value in the full flexion phase [15]. The average of the FRR of all of the channels was used as the FRR in the statistical analyses.

For the characterization of the variability of muscle activity distribution, the centroid coordinates of the RMS map (y-axis coordinates for the cranial-caudal direction) were extracted from the EMG signals [19]. We defined variability as the standard deviation (SD) of the centroid in each phase (standing, flexion phase, full flexion phase, extension phase). The average of the left and right variabilities of the three trials was used as the variability in the statistical analyses.

## Statistical analyses

We performed an association rule analysis to assess the association of each variable of pain-related factors and muscle activity (i.e., the FRR and the muscle variability) with CLBP disability (the RMDQ scores), and their complex associations. Association rules are used to uncover relationships between variables in transaction databases [47]. Analyses based on association rule mining have been conducted on a wide variety of datasets. The goal of this analysis is to identify patterns and combinations that meet a minimum requirement for prevalence and at

the same time occur much more frequently together than would be expected under statistical independence.

There are a number of terms used in association rules analyses. An 'association rule' is a relationship between sets of items ({A}→{B}) with an 'antecedent' {A} and a 'consequent' {B}. An 'item set' refers to a group of items. The term 'support' refers to the frequency of a particular item set. The 'confidence' of an association rule refers to how frequently an item occurs, conditional on an index item or item set. The 'lift' is a measure of the interestingness of an association rule that refers to how much more frequently two sets of items occur together compared to how often would be expected under statistical independence. Finally, 'interestingness' is a broad term that refers to items that are found to be interesting as a result of combinations of their support, confidence, lift, or other statistic.

When selecting association rules, constraints on various measures of significance and interest are useful. The best-known constraints are minimum thresholds of support, confidence, and lift. The support value of a rule with an antecedent item set A and a consequent item set B is defined as the proportion of transactions that include all antecedent and consequent item sets. In other words, support indicates the appearance rate of the rule. The support is measured as:

$$\text{Support}(A \rightarrow B) = p(A \cup B)/N$$

where p indicates probability and N represents the total number of transactions. Confidence is defined as the ratio of the support value to the number of transactions of all of the antecedent items sets; that is, confidence indicates the accuracy of the rule. The confidence is measured as:

$$\text{Confidence}(A \rightarrow B) = p(A \cup B)/p(A)$$

The lift value of a rule is the ratio of the number of transactions of consequent item sets (given that the antecedent item set has occurred) to the number of transactions of consequent item sets in all transactions. The lift is measured as:

$$\text{Lift}(A \rightarrow B) = p(A \cup B)/p(A)p(B)$$

A lift value >1 implies that the degree of association between the antecedent and consequent item sets is higher than in a situation in which the antecedent and consequent item sets are independent. We also used Fisher's exact test to assess the extracted association rule's generalizability [48]. Fisher's exact test can also be used for small sample sizes to compute exact nominal p-values.

Discretization is generally applied before an association rules analysis. We discretized each variable to binary format by using equal frequency discretization (EFD) as one of the discretization methods [49]. Each binarized variable was classified into two groups, a 'good' group and a 'poor' group. Each variable was compared between the two groups by the Mann-Whitney U-test, Student's t-test, or the Welch t-test. These tests were used according to the normality results from the Shapiro-Wilk tests. In the statistical analyses, the average of the FRR values of all of the channels was used as the FRR, and the average of the left and right variabilities of the three trials was used as the variability. Low values of the FRR, variability, and the PSEQ, and high values of other variables were classified as the poor group. For the confirmation of the influence of the angle on the FRR and the variability, we compared the maximum flexion angle of the trunk between the good and poor FRR groups and between the good and poor variability groups.

When we extracted rules with a consequence of poor RMDQ, the 'good' group was set to 0 and the 'poor' group was set to 1 as a dummy variable; we then performed an association rules analysis. To reduce and simplify the analysis, we used minimum thresholds that define interestingness as a lift value >1.0 [50]. This means that for all interesting rules ({A}→{B}) presented here, the joint set {A, B} occurs at least two times more frequently than we would expect under statistical independence. To extract rules, we set only one variable in the antecedent. When we extracted rules with a consequence of good RMDQ, the poor group was set to 0 and the good group was set to 1 as a dummy variable, and then we performed an association rules analysis. As conditions for extracting rules, we set only one variable in the antecedent, and we used minimum thresholds with a lift value >1.0.

To extract complex relevance, we set two variables in the antecedent, and we used minimum thresholds that define interestingness as those with confidence ≥80% [51]. This means that for all interesting rules ({A}→{B}) presented here, the consequent ({B}) occurred in ≥80% of all cases that showed morbidity in the antecedent ({A}). We then performed an association rules analysis. After extracting the association rules, we performed a hierarchical cluster analysis to divide the extracted association rules into some classes [52]. The statistical analyses were performed with R, ver. 3.6.1. The level of significance was set at $p < 0.05$.

## Results

### Pain-related factors

The characteristics of the 24 patients with CLBP lasting >3 months are summarized in Table 1. All pain-related factors were significantly different between the good group and the poor group ($p < 0.001$).

**Table 1. Baseline values of each variable's 'poor' and 'good' groups.**

| Variable | Total | Poor group | Good group | p value |
|---|---|---|---|---|
| Age | 71.6 ± 9.2 (Men: 71.5 ± 6.6, Women: 71.7 ± 10.7) | - | - | - |
| Gender | Men: Women | - | - | - |
| | 10:14 | | | |
| Pain duration (month) | 27.6 ± 43.9 | - | - | - |
| Trunk flexion angle (degrees) | 67.7 ± 11.9 | - | - | - |
| NRS pain | 4.3 ± 1.9 | 6.2 ± 1.2 (n = 9) | 3.0 ± 0.9 (n = 15) | p < 0.001 |
| SFMPQ-2 | 28.3 ± 22.2 | 44.8 ± 19.6 (n = 12) | 11.8 ± 7.3 (n = 12) | p < 0.001 |
| NRS fear | 3.5 ± 2.4 | 6.0 ± 1.7 (n = 9) | 2.0 ± 1.1 (n = 15) | p < 0.001 |
| PCS-4 | 6.7 ± 2.4 | 8.8 ± 0.9 (n = 10) | 5.1 ± 1.9 (n = 14) | p < 0.001 |
| HADS-anxiety | 5.0 ± 3.9 | 8.3 ± 3.0 (n = 11) | 2.2 ± 1.7 (n = 13) | p < 0.001 |
| HADS-depression | 6.2 ± 3.6 | 9.1 ± 2.7 (n = 12) | 3.3 ± 1.5 (n = 12) | p < 0.001 |
| PSEQ-2 | 8.2 ± 2.4 | 6.5 ± 1.8 (n = 13) | 10.3 ± 1.2 (n = 11) | p < 0.001 |
| TSK-11 | 14.4 ± 5.8 | 19.9 ± 4.2 (n = 9) | 11.1 ± 3.6 (n = 15) | p < 0.001 |
| CSI-9 | 13.8 ± 6.9 | 20.0 ± 5.7 (n = 10) | 9.4 ± 3.3 (n = 14) | p < 0.001 |
| FreBAQ | 10.2 ± 6.3 | 14.9 ± 5.3 (n = 12) | 5.4 ± 2.7 (n = 12) | p < 0.001 |
| RMDQ | 6.4 ± 4.3 | 10.0 ± 3.3 (n = 11) | 3.4 ± 1.9 (n = 13) | p < 0.001 |
| FRR | 12.1 ± 9.7 | 5.3 ± 1.2 (n = 12) | 18.9 ± 9.8 (n = 12) | p < 0.001 |
| Variability | 0.14 ± 0.07 | 0.09 ± 0.03 (n = 12) | 0.19 ± 0.06 (n = 12) | p < 0.001 |

Values are mean ± SD. CSI-9: Central Sensitization Inventory-9, FreBAQ: Fremantle Back Awareness Questionnaire, FRR: flexion relaxation ratio, HADS: Hospital Anxiety and Depression Scale, NRS: numerical rating scale, PCS-4: Pain Catastrophizing Scale-4, PSEQ-2: Pain Self Efficacy Questionnaire-2, RMDQ: Rolland-Morris Disability Questionnaire, SFMPQ-2: Short-Form McGill Pain Quesionnaire-2, TSK-11: Tampa Scale for Kinesiophobia-11.

## Electromyography

The FRR was significantly lower in the poor group compared to the good group (p<0.001), as was the variability (p<0.001). The RMS of each phase and the centroid of each phase are shown in Figs 3 and 4.

## Motion analysis

In the comparison of the maximum flexion angle of the trunk between the good FRR group (67.9 ± 11.8) and the poor FRR group (67.6 ± 13.2), the maximum flexion angle of the trunk was not significantly different (p = 0.96). Similarly, in the comparison of the maximum flexion angle of the trunk between the good variability group (71.0 ± 8.0) and the poor variability group (64.5 ± 15.0), the maximum flexion angle of trunk was not significantly different (p = 0.22).

## Association rules analysis

We fixed the lift values at >1.0, and the association rules had to have at least one poor pain-related factor or poor abnormal muscle activity in the antecedent and poor CLBP disability in the consequent. The clinically relevant rules are presented in Table 2. The FRR, TSK, NRS-fear, HADS-depression, FreBAQ, CSI, variability, SFMPQ, HADS-anxiety, NRS-intensity, and PCS were extracted in descending order of lift value for the association rules of high RMDQ (disability score). Based on the results of Fisher's exact test, the NRS for intensity as an associated rule and PCS as an associated rule were not acceptable. The highest lift was at low FRR (lift = 1.64). This meant that a low FRR and high RMDQ were highly associated, with a lift of 1.64, indicating that there was a 1.64-times higher likelihood of these two conditions occurring together than high RMDQ in isolation.

Table 3 shows the support, confidence, and lift values of the association rules between each pain-related factor or abnormal muscle activity in the antecedent and good CLBP disability in the consequent. We fixed the lift value at >1.0. The FRR, HADS-depression, FreBAQ, variability, SFMPQ2, TSK, NRS-fear, CSI, HADS-anxiety, NRS-intensity, and PCS were extracted in

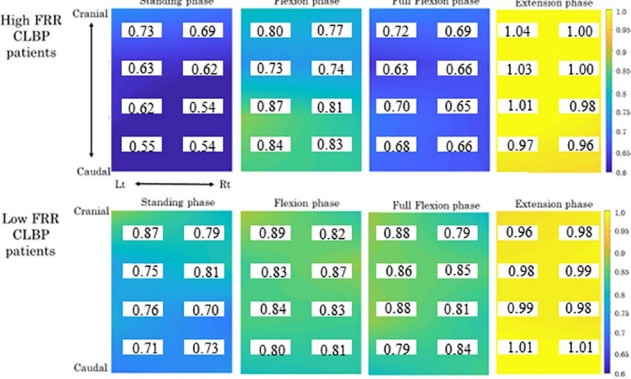

**Fig 3. The differences in the topographical maps of the RMS value between the low-FRR and high-FRR groups.**
Representative topographical maps of the electromyographic RMS values recorded from the bilateral erector spinae for a high-FRR patient and a low-FRR patient in the standing phase, flexion phase, full flexion phase, and extension phase of the standing forward bending task. Blue indicates low muscle activity, and yellow indicates high muscle activity. The RMS values of the low-FRR patient were lower than those of the high-FRR patients in the standing phase and full flexion phase. The RMS value of each channel was insert in topographical maps. The changes in the RMS value between the flexion phase and the full flexion phase in the low-FRR patients were smaller compared to those of the high-FRR patients.

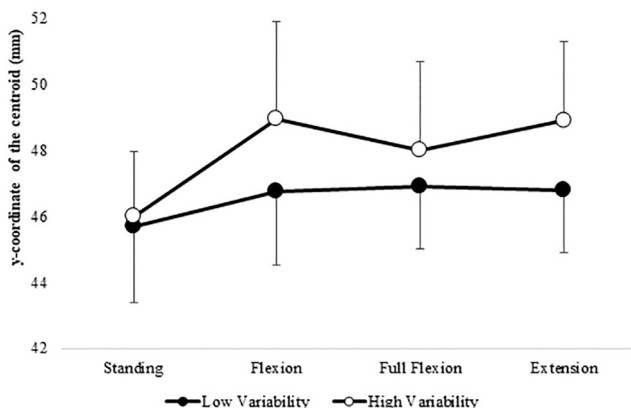

**Fig 4. The difference in the y-coordinate of the centroid between the low-variability and high-variability groups.** The mean (±SD) of the y-axis coordinate of the centroid of the RMS map estimated in the standing phase, flexion phase, full flexion phase, and extension phase of the standing forward bending task. The high-variability group tended to show greater changes in the y-coordinate of the centroid compared to the low-variability group.

descending order of lift value for the association rules of low RMDQ (disability score). Based on the results of Fisher's exact test, the NRS for intensity as an associated rule and PCS as an associated rule were not acceptable. The highest lift was at high FRR (lift = 1.54), which means that high FRR and low RMDQ were highly associated, with a lift of 1.54, indicating that these two conditions were very closely linked and the probability of them occurring together was 1.54-times higher than low RMDQ in isolation.

We fixed the minimum confidence at 80% and the lift values at >1.0, and the association rules had to have at least one or two poor pain-related factors and poor abnormal muscle activity in the antecedent and poor CLBP disability in the consequent. The clinically relevant rules are presented in Table 4. A total of 21 association rules were found to be clinically relevant. Fisher's exact test revealed that all 21 association rules were acceptable. In the results of the cluster analysis, the 21 rules of severe CLBP disability were divisible into 'low FRR-related rules,' 'high HADS depression-related rules,' 'high TSK and NRS fear-related rules,' 'high CSI-related

**Table 2. Association rules sorted by lift value (Consequent = RMDQ high).**

| No | Antecedent ⇒ Consequent | | Confidence | Support | Lift | Fisher's exact value |
|----|-------------------------|----------------|-----------|---------|------|----------------------|
| 1 | FRR Low | ⇒RMDQ High | 75% | 38% | 1.64 | < 0.001 |
| 2 | TSK High | ⇒RMDQ High | 67% | 25% | 1.45 | < 0.001 |
| 3 | NRS fear High | ⇒RMDQ High | 67% | 25% | 1.45 | < 0.001 |
| 4 | HADS-depression High | ⇒RMDQ High | 67% | 33% | 1.45 | < 0.001 |
| 5 | FreBAQ High | ⇒RMDQ High | 67% | 33% | 1.45 | < 0.001 |
| 6 | CSI High | ⇒RMDQ High | 60% | 25% | 1.31 | < 0.001 |
| 7 | Variability Low | ⇒RMDQ High | 58% | 29% | 1.27 | < 0.001 |
| 8 | SFMPQ-2 High | ⇒RMDQ High | 58% | 29% | 1.27 | < 0.001 |
| 9 | HADS-anxiety High | ⇒RMDQ High | 55% | 25% | 1.19 | < 0.001 |
| 10 | NRS intensity High | ⇒RMDQ High | 50% | 21% | 1.09 | 0.13 |
| 11 | PCS High | ⇒RMDQ High | 50% | 21% | 1.09 | 0.13 |

Abbreviations are explained in the Table 1 footnote.

**Table 3. Association rules sorted by lift values (Consequent = RMDQ low).**

| No | Antecedent ⇒ Consequent | | Confidence | Support | Lift | Fisher's exact value |
|---|---|---|---|---|---|---|
| 1 | FRR High | ⇒RMDQ Low | 83% | 42% | 1.54 | < 0.001 |
| 2 | HADS-depression Low | ⇒RMDQ Low | 75% | 38% | 1.38 | < 0.001 |
| 3 | FreBAQ Low | ⇒RMDQ Low | 75% | 38% | 1.38 | < 0.001 |
| 4 | Variability High | ⇒RMDQ Low | 67% | 33% | 1.23 | < 0.001 |
| 5 | SFMPQ-2 Low | ⇒RMDQ Low | 67% | 33% | 1.23 | < 0.001 |
| 6 | TSK Low | ⇒RMDQ Low | 67% | 42% | 1.23 | < 0.001 |
| 7 | NRS fear Low | ⇒RMDQ Low | 67% | 42% | 1.23 | < 0.001 |
| 8 | CSI Low | ⇒RMDQ Low | 64% | 38% | 1.19 | < 0.001 |
| 9 | HADS-anxiety Low | ⇒RMDQ Low | 62% | 33% | 1.14 | < 0.001 |
| 10 | NRS intensity Low | ⇒RMDQ Low | 57% | 33% | 1.05 | 0.13 |
| 11 | PCS Low | ⇒RMDQ Low | 57% | 33% | 1.05 | 0.13 |

Abbreviations are explained in the Table 1 footnote.

rules,' and 'high PCS-related rules.' For example, the low FRR-related rules indicate that low FRR is associated with all rules. These clustering association rules are visualized in Fig 5.

Table 5 provides the support, confidence, and lift values of the association rules between one or two pain-related factors and abnormal muscle activity in antecedent and good CLBP disability in consequent. We fixed the minimum confidence at 80% and the lift values at >1.0. A total of 23 association rules were found to be clinically relevant. According to the Fisher's

**Table 4. Association rules: Antecedent = included two variables, consequent = RMDQ high, minimum confidence = 80%.**

| No | Antecedent ⇒ Consequent | | Confidence | Support | Lift | Fisher's exact value |
|---|---|---|---|---|---|---|
| 1 | FRRLow and SFMPQ-2 High | ⇒RMDQ High | 100% | 21% | 2.18 | < 0.001 |
| 2 | FRRLow and NRS intensity High | ⇒RMDQ High | 100% | 17% | 2.18 | < 0.001 |
| 3 | FRRLow and NRS fear High | ⇒RMDQ High | 100% | 17% | 2.18 | < 0.001 |
| 4 | FRRLow and TSK High | ⇒RMDQ High | 100% | 21% | 2.18 | < 0.001 |
| 5 | FRR Low and PSEQ Low | ⇒RMDQ High | 80% | 17% | 1.75 | < 0.001 |
| 6 | FRRLow and HADS-depression High | ⇒RMDQ High | 88% | 29% | 1.91 | < 0.001 |
| 7 | FRRLow and FreBAQ High | ⇒RMDQ High | 88% | 29% | 1.91 | < 0.001 |
| 8 | HADS-depression High and Variability Low | ⇒RMDQ High | 88% | 29% | 1.91 | < 0.001 |
| 9 | HADS-depression High and FreBAQ High | ⇒RMDQ High | 88% | 29% | 1.91 | < 0.001 |
| 10 | HADS-depression High and CSI High | ⇒RMDQ High | 80% | 17% | 1.75 | < 0.001 |
| 11 | HADS-depression High and PCS High | ⇒RMDQ High | 80% | 17% | 1.75 | < 0.001 |
| 12 | TSK High and NRS intensity High | ⇒RMDQ High | 83% | 21% | 1.82 | < 0.001 |
| 13 | TSK High and Variability Low | ⇒RMDQ High | 80% | 17% | 1.75 | < 0.001 |
| 14 | TSK High and FreBAQ High | ⇒RMDQ High | 80% | 17% | 1.75 | < 0.001 |
| 15 | NRS fear High and FreBAQ High | ⇒RMDQ High | 80% | 17% | 1.75 | < 0.001 |
| 16 | NRS fear High and NRS intensity High | ⇒RMDQ High | 80% | 17% | 1.75 | < 0.001 |
| 17 | CSI High and SFMPQ-2 High | ⇒RMDQ High | 86% | 25% | 1.87 | < 0.001 |
| 18 | CSI High and NRS fear High | ⇒RMDQ High | 83% | 21% | 1.82 | < 0.001 |
| 19 | CSI High and TSK High | ⇒RMDQ High | 80% | 17% | 1.75 | < 0.001 |
| 20 | PCS High and NRS fear High | ⇒RMDQ High | 83% | 21% | 1.82 | < 0.001 |
| 21 | PCS High and SFMPQ-2 High | ⇒RMDQ High | 83% | 21% | 1.82 | < 0.001 |

Abbreviations are explained in the Table 1 footnote.

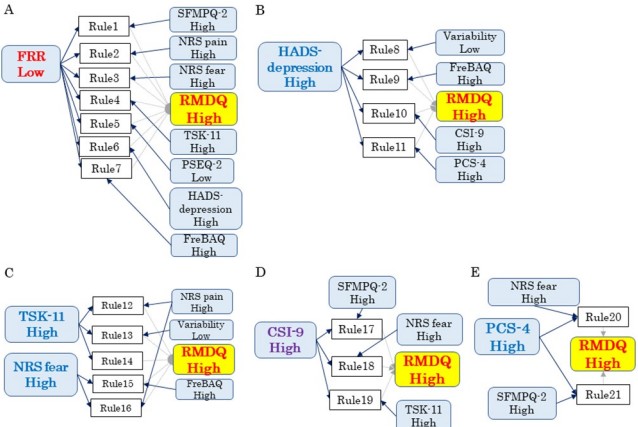

**Fig 5. Association rules: A graphic view related to high RMDQ scores.** The rules of severe CLBP disability were divisible into (**A**) low FRR-related rules, (**B**) high HADS depression-related rules, (**C**) high TSK and NRS fear-related rules, (**D**) high CSI-related rules, and (**E**) high PCS-related rules.

exact test results, all association rules were acceptable. The cluster analysis results showed that the 23 rules of good CLBP disability were divisible into 'high FRR-related rules,' 'low FreBAQ-related rules', 'low HADS depression-related rules,' and 'low muscle variability-related rules.' These clustering association rules are illustrated in Fig 6. When we combined pain-related

**Table 5. Association rules: Antecedent = included two variables, consequent = RMDQ low, minimum confidence = 80%.**

| No | Antecedent ⇒ Consequent | | Confidence | Support | Lift | Fisher's exact value |
|---|---|---|---|---|---|---|
| 1 | FRR High and CSI Low | ⇒RMDQ Low | 100% | 33% | 1.85 | < 0.001 |
| 2 | FRR High and NRS fear Low | ⇒RMDQ Low | 100% | 29% | 1.85 | < 0.001 |
| 3 | FRR High and PCS Low | ⇒RMDQ Low | 100% | 29% | 1.85 | < 0.001 |
| 4 | FRR High and SFMPQ-2 Low | ⇒RMDQ Low | 100% | 21% | 1.85 | < 0.001 |
| 5 | FRR High and NRS intensity Low | ⇒RMDQ Low | 83% | 21% | 1.54 | < 0.001 |
| 6 | FRR High and TSK Low | ⇒RMDQ Low | 88% | 29% | 1.62 | < 0.001 |
| 7 | FRR High and HADS-anxiety Low | ⇒RMDQ Low | 88% | 29% | 1.62 | < 0.001 |
| 8 | FRR High and HADS-depression Low | ⇒RMDQ Low | 88% | 29% | 1.62 | < 0.001 |
| 9 | FRR High and FreBAQ Low | ⇒RMDQ Low | 88% | 29% | 1.62 | < 0.001 |
| 10 | FRR High and Variability High | ⇒RMDQ Low | 88% | 29% | 1.62 | < 0.001 |
| 11 | FRR High | ⇒RMDQ Low | 83% | 42% | 1.54 | < 0.001 |
| 12 | FreBAQ Low and NRS intensity Low | ⇒RMDQ Low | 88% | 29% | 1.62 | < 0.001 |
| 13 | FreBAQ Low and NRS fear Low | ⇒RMDQ Low | 88% | 29% | 1.62 | < 0.001 |
| 14 | FreBAQ Low and TSK Low | ⇒RMDQ Low | 88% | 29% | 1.62 | < 0.001 |
| 15 | FreBAQ Low and PSEQ High | ⇒RMDQ Low | 83% | 21% | 1.54 | < 0.001 |
| 16 | FreBAQ Low and CSI Low | ⇒RMDQ Low | 80% | 33% | 1.48 | < 0.001 |
| 17 | HADS-depression Low and TSK Low | ⇒RMDQ Low | 89% | 33% | 1.64 | < 0.001 |
| 18 | HADS-depression Low and SFMPQ-2 Low | ⇒RMDQ Low | 88% | 29% | 1.62 | < 0.001 |
| 19 | HADS-depression Low and CSI Low | ⇒RMDQ Low | 86% | 25% | 1.58 | < 0.001 |
| 20 | HADS-depression Low and NRS intensity Low | ⇒RMDQ Low | 86% | 25% | 1.58 | < 0.001 |
| 21 | HADS-depression Low and NRS fear Low | ⇒RMDQ Low | 80% | 33% | 1.48 | < 0.001 |
| 22 | Variability High and SFMPQ-2 Low | ⇒RMDQ Low | 83% | 21% | 1.54 | < 0.001 |
| 23 | Variability High and PSEQ High | ⇒RMDQ Low | 80% | 17% | 1.48 | < 0.001 |

Abbreviations are explained in the Table 1 footnote.

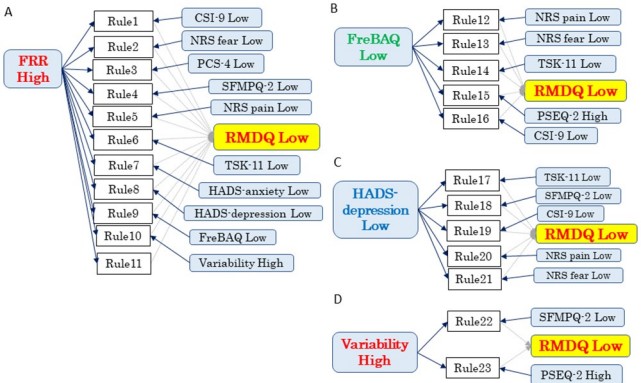

**Fig 6. Association rules: A graphic view related to low RMDQ.** The rules of mild CLBP disability were divisible into (**A**) high FRR-related rules, (**B**) low FreBAQ-related rules, (**C**) low HADS depression-related rules, and (**D**) high variability-related rules.

factors with the FRR and muscle variability, the relationships between FRR/muscle variability and CLBP disability became stronger, which indicated an increase in the lift value.

## Discussion

We used association rules analyses to investigate the relationships among CLBP disability, muscle activity, and pain-related factors in 24 patients with CLBP. The results demonstrated that a reduction of the FRR and muscle variability were associated with CLBP disability. In particular, the FRR was most strongly associated with CLBP disability. We also performed a cluster analysis of association rules to investigate the complex relationships among CLBP disability, muscle activity, and pain-related factors. The rules of severe CLBP disability were categorized into 'severe reduction of FRR-related rules,' 'severe depression-related rules,' 'severe kinesiophobia and fear about lumbar movement-related rules,' 'severe central sensitivity syndromes-related rules,' and 'severe catastrophizing-related rules.' In contrast, the rules of mild CLBP disability were divisible into 'mild reduction of FRR-related rules,' 'mild body perception disturbance-related rules,' 'mild depression-related rules,' and 'mild reduction of muscle variability-related rules.' When pain-related factors were combined with the FRR and muscle variability, the close relationships between FRR/muscle variability and CLBP disability became stronger. Therefore, abnormal muscle activity was more strongly associated with CLBP disability when it was combined with pain-related factors.

Several cross-sectional studies showed that the FRR [26–28], pain severity [4], psychological factors [4], body perception disturbance [12], and central sensitivity syndromes [13] were associated with CLBP disability. However, these studies did not compare the degree of contribution of each factor, including the FRR and pain-related factors, to CLBP disability. In the present study, the FRR was identified as the factor with the greatest effect on CLBP disability compared to the other pain-related factors. In the previous studies it was unclear how the combination of FRR and pain-related factors could affect CLBP disability. Our present findings, based on the cluster analysis of the association rules, demonstrated that CLBP disability was more affected by the FRR than by other pain-related factors. This relationship between the FRR and CLBP disability became stronger when combined with other pain-related factors. Our results thus emphasize the importance of intervention for an FRR that is causing motor control dysfunction. Other pain-related factors (pain, fear, depression, self-efficacy, and body

perception) should also be considered, as these factors might facilitate a reduction of the FRR or contribute to CLBP disability.

Regarding the variability of muscle activity distribution, a cross-sectional study showed that low muscle variability was associated with CLBP disability [30], but that study did not compare the contribution of each variable (including muscle variability and pain-related factors) to CLBP disability. Herein we observed that the impact of muscle variability on CLBP disability was not higher than that of the FRR, but muscle variability was extracted as a factor with a high probability of affecting CLBP disability. In the earlier study [30], it was also not certain how the combination of muscle variability and pain-related factors would affect CLBP disability. According to the results of our present cluster analysis of association rules, muscle variability was associated with CLBP disability in combination with pain severity and self-efficacy. As a result, the reduction of muscle variability affected the patients' CLBP disability, and muscle variability in combination with other pain-related factors was associated with CLBP disability. For that reason, our findings highlight the need for interventions for the motor control dysfunction that causes the reduction of muscle variability in CLBP patients. In addition, pain severity and self-efficacy might affect CLBP disability as facilitators of the reduction in muscle variability or by contributing to CLBP disability. Pain severity in particular was reportedly associated with changes in motor variability [53]. No studies have been reported to date to examine whether the intervention can improve muscle activity variability reduction. However, EMG feedback therapy with high-density EMG is a potential future intervention [54]. Based on the association between pain and muscle activity, a decrease in pain and lumbar muscle stiffness has been reported after spinal manipulation [55]. Interventions that focus on muscle activity or on factors affecting muscle activity may be effective in improving muscle activity distribution variability.

The present study is the first to demonstrate the complex associations among abnormal muscle activity (as an indicator of motor control dysfunction), CLBP disability, and pain-related factors in CLBP patients by using an association rules analysis. Our results could be clinically plausible in light of the contemporary theory of motor adaptation to pain [56], which proposes that pain and pain-related factors cause motor changes through the sensorimotor system, resulting in behavioral changes. Our findings thus suggest that changes in motor control such as a reduction of the FRR and the variability of muscle activity distribution lead to CLBP disability. In particular, a low FRR was more strongly associated with CLBP disability than other pain-related factors in this study. These results may indicate that the reduction of the FRR was induced by pain and pain-related factors through the sensorimotor system, and this directly affected the patients' CLBP disability. Physical therapy such as motor control exercises [57], spinal manipulation [58], and biofeedback training [59] to improve a low FRR could help reduce CLBP disability.

An earlier study reported that the reduction of FRR and CLBP disability are longitudinally linked [58]. Mechanisms underlying a reduction of the FRR have been proposed: mechanical factors such as a reduced range of lumbar movement [60], muscle guarding due to pain [61], and altered neuromuscular co-ordination between the trunk and hip [62]; thus, increasing efforts to enhance spinal stability could be effective [63]. Psychological factors such as fear were also thought to contribute to the FRR [23]. Herein, we classified the rules related to CLBP disability into FRR-related rules, and the FRR-related rules were classified separately from psychological factor-related rules; thus, the FRR and psychological factors may be considered as different categories. However, when we combined pain-related factors with the FRR, the relationship between the FRR and CLBP disability became stronger, suggesting the need for comprehensive interventions that include both physical exercise and psychological factors. Such comprehensive interventions may lead to much greater improvements of CLBP disability.

Further research is necessary to investigate how such interventions affect the FRR, pain severity, psychological factors, body perception disturbance, and central sensitization syndrome longitudinally.

This study had several limitations. (1) The sample size of this study is not large. A small sample size increases the probability that one patient's results will be affected by small sample size in association rule analysis (especially "support"). In the present study, the Fisher exact test was performed, one method of examining generalizability when the sample size is small. However, the current study results did not exclude the possibility that the sample size affected the results. (2) The small sample size, the type of LBP experienced by participants, and the specific assessment techniques used limit the generalizability of this study's findings. (3) The standing trunk flexion and re-extension task was performed by the patients for the calculation of the FRR, and although the time was set in each movement phase, the velocity of the patients' movements could not be finely controlled. For this reason, individual differences in the fine velocity may have influenced the muscle activity. (4) the outcomes of pain-related factors were all assessed using questionnaires, which might result in questionnaire bias. (5) We could not determine the mechanisms underlying the relationships among abnormal FRR/muscle variability and pain-related factors and CLBP disability because we did not measure the activity of central nervous system. (6) Differences in the LBP site might affect the muscle activity values. There were variations in the patients' LBP sites, and there were some with and without LBP at the EMG electrode application site. However, none of the patients reported experiencing any pain during the standing trunk flexion re-extension task, and perhaps at least movement pain did not affect muscle activity. (7) The current study was a cross-sectional study, and it was not possible to design an intervention based on a longitudinal course. Developing an intervention based on a longitudinal time is necessary for future studies.

To our knowledge, this is the first study to investigate the complex associations among CLBP disability, abnormal muscle activity, and pain-related factors in CLBP patients by conducting association rules analyses. When pain-related factors were combined with the FRR and muscle variability, the relationships between abnormal FRR/muscle variability and CLBP disability became stronger. Our results suggest the importance of focusing on not only the pain-related factors in CLBP patients but also their abnormal motor control, which causes CLBP disability. Our findings could help clinical design interventions, which may improve the disability of numerous individuals with CLBP.

## Supporting information

**S1 File. Supporting information file.**
(XLSX)

## Author Contributions

**Conceptualization:** Hayato Shigetoh.

**Data curation:** Hayato Shigetoh.

**Formal analysis:** Hayato Shigetoh, Yuki Nishi, Michihiro Osumi.

**Investigation:** Hayato Shigetoh.

**Methodology:** Hayato Shigetoh, Yuki Nishi, Michihiro Osumi.

**Project administration:** Hayato Shigetoh.

**Supervision:** Shu Morioka.

**Visualization:** Hayato Shigetoh, Michihiro Osumi.

**Writing – original draft:** Hayato Shigetoh.

**Writing – review & editing:** Michihiro Osumi, Shu Morioka.

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
