## [Decision Letter · Decision Letter 0]

2 Nov 2020

PONE-D-20-19882

Combined abnormal muscle activity and pain-related factors affect disability in patients with chronic low back pain: An association rule analysis

PLOS ONE

Dear Dr. Shigetoh,

Thank you for submitting your manuscript to PLOS ONE. After careful consideration, we feel that it has merit but does not fully meet PLOS ONE’s publication criteria as it currently stands. Therefore, we invite you to submit a revised version of the manuscript that addresses the points raised during the review process.

We look forward to receiving your revised manuscript.

Kind regards,

Chris Connaboy

Academic Editor

PLOS ONE

Journal Requirements:

2. Please state in your methods section when you conducted this study.

3. Please provide details of the obtained participant consent in the ethics statement on the online submission form. Currently this information is only available in the methods section of your manuscript.

Reviewers' comments:

Reviewer's Responses to Questions

**Comments to the Author**

1. Is the manuscript technically sound, and do the data support the conclusions?

Reviewer #1: Partly

Reviewer #2: Yes

2. Has the statistical analysis been performed appropriately and rigorously? 

Reviewer #1: Yes

Reviewer #2: Yes

3. Have the authors made all data underlying the findings in their manuscript fully available?

Reviewer #1: No

Reviewer #2: Yes

4. Is the manuscript presented in an intelligible fashion and written in standard English?

Reviewer #1: Yes

Reviewer #2: Yes

5. Review Comments to the Author

Reviewer #1: Thank you for the opportunity to review the manuscript. The study aims to investigate the relationship between a number of pain/disability related variables and EMG measures of the lower back during a standing flexion and extension task in 24 adults with chronic low back pain. These links are explored with a complex set of analyses, including an association rule analysis.

The manuscript is well written and the methods described in adequate details for the most part. Please find below my major and minor points for feedback.

Major:

-The principle concern I have with this study is that it is not clear how the sample size was determined and whether, given the number and nature of analyses performed, if just 24 participants is sufficient. Can the authors please justify their sample size? If a priori calculation was not performed, the authors could consider a post hoc calculation.

-There appears to be insufficient detail about the LBP reported by participants. Can the authors please included (if available), the following: location of LBP, and further detail about the nature of the chronic LBP. The methods outline some exclusion criteria regarding other conditions but can the authors please clarify if the LBP is nonspecific or due to a specific cause? Please see the following article (doi provided) for recommendation on relevant criteria to report: 10.1186/s12891-018-2034-6

Minor:

Introduction

-Page 5, line 69-71: please provide references to support the following sentence "In CLBP patients, the FRR is reportedly decreased, indicating that they cannot relax their spinal extensor muscles at the end range of lumbo-pelvic flexion."

-Page 5, line 74-76: please be specific with the regions of muscle activation that were restricted in studies referenced.

Methods

-Page 6, line 97: As per my second major point above, please clarify if patients had specific LBP, nonspecific LBP, or mixed throughout the cohort.

-Page 10, line 160: Please replace 'record' with 'recording' for the subheading.

-Page 10, line 163: Figure 2 does not clearly indicate the location of the accelerometer, please update.

-Page 11, line 179: Besides the use of an alcohol wipe, was skin abraded or hair removed?

-Page 11, line 182: Please provide a reference or justification for the placement of the EMG electrodes, e.g. SENIAM guidelines.

-Were data assessed for normality? Please clarify and justify approach.

Results

-Participants appear to be sub-grouped into 'poor' and 'good' for each variable. Please clarify the number of participants in each group

-Figure 3 is not particularly clear (image quality) and I'm not sure of the clinical relevance of the topographical map, can the authors justify this approach, please?

-Can the values from the EMG measures be reported, please?

-Page 18, line 302: RMS has been previously defined. Please be consistent throughout - I don't think it is necessary to re-define at this point.

Discussion

-Page 28, line 407-408: There appears to be a word/s missing after the second comma. Perhaps add 'demonstrated that'

-Page 28, line 412: Please remove the second use of the word 'be' so that it reads as '...factors might facilitate...". Additionally, please specify the ‘other pain-related variables’ that you mean.

-Page 29, line 427: Please suggest some examples of interventions that might provide a reduction in the variability of muscle activity in people with CLBP.

-Page 30, line 443: I am not convinced that these interventions, or the findings from your study, justify this recommended. Please remove the words 'thus resolve' and replace with 'help reduce'

-Page 31, line 456: Please replace the word 'should' with 'may'

-Limitations: the sample size, multiple statistical analyses, and sub-grouping of participants are a concern and should be addressed as per the major feedback points above. Additionally, please consider adding the following as limitations: generalisability of results (based on methods e.g. FRR and participant characteristics).

-Page 32, line 478: Please explain how the results of the study improve the disability of with CLBP? I don't believe any interventions were tested. Please consider rephrasing to relate to understanding of function/relationship between variables measured, or something similar.

-Page 32, line 479: Please abbreviate chronic low back pain, as per previous sections of the manuscript.

Reviewer #2: In general, the article is well versed, efficiently organized, translated the knowledge in elaborative manner and highlighted the dark zones in the muscle activity and pain-related areas using sEMG and open new horizons for researchers in this field. Although, this study is methodologically sound. However, the progression on previous work is not substantial, and some specific concerns are listed below.

1.The introduction (rationale) need to be strengthen to highlight the significance of the present investigation. Like, there was little information about surface EMG, while this is the key part of the manuscript.

2.Page 11, Line 79-80: Can you split the sentences, “Several research groups have…”. Write each group’s individual work(s).

3.Page 12, Line: 97: “Patients”: include demographic characteristics of men and women separately (age, weight, height etc.). And instead of male and female, change to men and women through the MS.

4.Page 16, section “Trunk kinematic record and analysis”: you did not mention about the any axis of the accelerometer (3-axis). How did you calculated the kinematic parameters? Has this device been validated before against lab standards? Change the figure 2 and put a real subject’s picture with accelerometer, also indicate the EMG grid and accelerometer. Line 162: Please change the sentence as “An accelerometer was fixed over the patient's twelfth thoracic (T12)…”

5.The EMG was recorded with 1000 Hz sampling freq, and accelerometer with 100 Hz. So, how do you combine both the signals? Or, you have extracted/analyzed separately?

6.Is only Normalized RMS from EMG adequate to get the successful results?

7.Page 20, line 244:: “Each binarized variable was classified into two groups, a 'good' group and a 'poor' group”, how do you scaled poor and good? Did you find any previous research work(s)?

6. PLOS authors have the option to publish the peer review history of their article (what does this mean?). If published, this will include your full peer review and any attached files.

Reviewer #1: No

Reviewer #2: No

---

## [Author Response · Author response to Decision Letter 0]

17 Nov 2020

15-November-2020

Dr. Joerg Heber

Editor-in-Chief

PLOS ONE

Dear Editors:

Manuscript ID PONE-D-20-19882, " Combined abnormal muscle activity and pain-related factors affect disability in patients with chronic low back pain: An association rule analysis," by Shigetoh et al. ' 

Please find attached a revised version of our above-named manuscript, which we would like to resubmit for publication as a Research Article in PLOS ONE.

Your comments and those of the reviewers were highly insightful and enabled us to greatly improve the quality of our manuscript. In the following pages are our point-by-point responses to each of the comments of the reviewers as well as your own comments.

Revisions in the text are shown using red-colored font. In accord with Reviewer No.1’s suggestion, we revised text in the Introduction, Methods, Results, Discussion and Figure. In accord with Reviewer No.2’s suggestion, we revised text in the Introduction, Methods, Results, Discussion, and Figure. We hope that the revisions in the manuscript and our accompanying responses will be sufficient to make our manuscript suitable for publication in PLOS ONE.

We look forward to hearing from you at your earliest convenience.

Sincerely,

Dr. Hayato Shigetoh

Department of Neurorehabilitation

Graduate School of Health Sciences, Kio University

4 Chome-2-2 Umaminaka, Koryo, Kitakatsuragi District, Nara 635-0832, Japan

Tel.: +81-745-54-1601

Fax: +81-745-54-1600

Email: hayato.pt1121@gmail.com

Manuscript ID PONE-D-20-19882, " Combined abnormal muscle activity and pain-related factors affect disability in patients with chronic low back pain: An association rule analysis," by Shigetoh et al. ' 

Responses to the comments of Reviewer #1

Reviewer #1: 

Thank you for the opportunity to review the manuscript. The study aims to investigate the relationship between a number of pain/disability related variables and EMG measures of the lower back during a standing flexion and extension task in 24 adults with chronic low back pain. These links are explored with a complex set of analyses, including an association rule analysis.

The manuscript is well written and the methods described in adequate details for the most part. Please find below my major and minor points for feedback.

Response: Thank you for reviewing our manuscript. Your comments were highly insightful and enabled us to greatly improve the quality of our manuscript. In the following pages are our point-by-point responses to each of the comments of the reviewers as well as your own comments. Revisions in the text are shown using red-colored font.

Major:

-The principle concern I have with this study is that it is not clear how the sample size was determined and whether, given the number and nature of analyses performed, if just 24 participants is sufficient. Can the authors please justify their sample size? If a priori calculation was not performed, the authors could consider a post hoc calculation.

Response: Association rule analysis was used in the present study. Since association rule analysis is based on probability, it was difficult to determine the sample size in advance and examine power as a post hoc analysis. However, as noted in the reviewer's comments, a small sample size also increases the probability that one patient's results will be affected (especially "support"). In the present study, the Fisher exact test was performed, one method of examining generalizability when the sample size is small. However, the current study results did not exclude the possibility that the sample size affected the results. We added the limitation. We revised the Discussion section. We added the following text:

Line 482: (1) The sample size of this study is not large. A small sample size increases the probability that one patient's results will be affected by small sample size in association rule analysis (epecially “support”). In the present study, the Fisher exact test was performed, one method of examining generalizability when the sample size is small. However, the current study results did not exclude the possibility that the sample size affected the results.

-There appears to be insufficient detail about the LBP reported by participants. Can the authors please included (if available), the following: location of LBP, and further detail about the nature of the chronic LBP. The methods outline some exclusion criteria regarding other conditions but can the authors please clarify if the LBP is nonspecific or due to a specific cause? Please see the following article (doi provided) for recommendation on relevant criteria to report: 10.1186/s12891-018-2034-6

Response: Thank you for recommendation article on relevant criteria. The recommended article was on the criteria for non-specific low back pain. Although this study was not limited to non-specific low back pain, we referred to the study's criteria section. We also added details of pain location and with or without referred leg pain, and specific and non-specific low back pain. We revised the Methods section. We added the following text:

Line 106: The patient's LBP area's detail was a low back area (n =12) and between sacroiliac joint and gluteal folds inferiorly (n =12). The patients had referred leg pain (n =2) and no referred leg pain (n =22). Also, the patient's LBP categorized to non-specific LBP (n = 6) and specific LBP (n = 18). The specific LBP was included spinal stenosis (n = 3) and lumbar osteoarthritis (n =15).

Minor:

Introduction

-Page 5, line 69-71: please provide references to support the following sentence "In CLBP patients, the FRR is reportedly decreased, indicating that they cannot relax their spinal extensor muscles at the end range of lumbo-pelvic flexion."

Response: We have done so. We revised the Introduction section. We added the following text:

Line 70: In CLBP patients, the FRR is reportedly decreased, indicating that they cannot relax their spinal extensor muscles at the end range of lumbo-pelvic flexion [15,17].

-Page 5, line 74-76: please be specific with the regions of muscle activation that were restricted in studies referenced.

Response: Previous studies have reported that crania regions were activated than caudal regions. We revised the Introduction section. We added the following text:

Line 75: These studies also indicated that the regions of muscle activation in CLBP patients were restricted and cranial regions were activated than caudal regions.

Methods

-Page 6, line 97: As per my second major point above, please clarify if patients had specific LBP, nonspecific LBP, or mixed throughout the cohort.

Response: In this study, we included specific low back pain and non-specific low back pain, which were mixed throughout the cohort. We revised the Methods section. We added the following text:

Line 108: Also, the patient's LBP categorized to non-specific LBP (n = 6) and specific LBP (n = 18). The specific LBP was included spinal stenosis (n = 3) and lumbar osteoarthritis (n =15).

-Page 10, line 160: Please replace 'record' with 'recording' for the subheading.

Response: We have done so. We revised the Methods section. We added the following text:

Line 170: Trunk kinematic recording and analysis

-Page 10, line 163: Figure 2 does not clearly indicate the location of the accelerometer, please update.

Response: We have done so. There was no illustration of where the accelerometer was placed. We revised the Figure 2. We added the illustration of the location of the accelerometer.

-Page 11, line 179: Besides the use of an alcohol wipe, was skin abraded or hair removed?

Response: We were also careful about skin abraded. Also, the patient`s skin over the erector spinae was no hair in all patients. We revised the Methods section. We added the following text:

Line 191: The patient's glabrous skin over the erector spinae was abraded skin and cleaned with alcohol.

-Page 11, line 182: Please provide a reference or justification for the placement of the EMG electrodes, e.g. SENIAM guidelines.

Response: We have followed previous studies [Reference No. 45] and decided the placement of the EMG electrodes. We have added citation. We revised the Methods and Reference section. We added the following text:

Line 192: Two grids of sheet-type electrodes coated with an electroconductive gel were taped to the skin at that site; the electrode was located 3 cm outside the midpoint of the lumbar spinous process [45] and covered upward along the erector spinae from the level of the Jacoby line (Fig. 2).

-Were data assessed for normality? Please clarify and justify approach.

Response: We have done so. We used the Shapiro-Wilk tests to assess for normality. We revised the Methods section. We added the following text:

Line 258: Each variable was compared between the two groups by the Mann-Whitney U-test, Student's t-test, or the Welch t-test. These tests were used according to the normality results from the Shapiro-Wilk tests.

Results

-Participants appear to be sub-grouped into 'poor' and 'good' for each variable. Please clarify the number of participants in each group

Response: We have done so. We added the number of participants in each group for each variable in Table 1. We revised the Table 1.

-Figure 3 is not particularly clear (image quality) and I'm not sure of the clinical relevance of the topographical map, can the authors justify this approach, please?

Response: Based on the report by Falla et al. (Reference No. 19), the color tone was changed and mapped according to the magnitude of muscle activity at each site. The color criterion is shown in Figure 3; blue indicates low muscle activity, and yellow indicates high muscle activity. Topographical maps make it easier to capture muscle activity and muscle activity distribution visually. We believe that their clinical relevance is related to the reduced variability of muscle activity and muscle activity distribution characteristics of patients with low back pain. We revised the Results section. We added the following text:

Line 309: Blue indicates low muscle activity, and yellow indicates high muscle activity.

-Can the values from the EMG measures be reported, please?

Response: We added the RMS values for each EMG channel and movement phase in the topographical maps in Figure 3. We revised the Figure 3.

-Page 18, line 302: RMS has been previously defined. Please be consistent throughout - I don't think it is necessary to re-define at this point.

Response: We have done so. We revised the Methods section. We revised “the root mean square (RMS)” to “the RMS”. (Line 318)

Discussion

-Page 28, line 407-408: There appears to be a word/s missing after the second comma. Perhaps add 'demonstrated that'

Response: We have done so. We revised the Discussion section. We added the following text:

Line 423: Our present findings, based on the cluster analysis of the association rules, demonstrated that CLBP disability was more affected by the FRR than by other pain-related factors.

-Page 28, line 412: Please remove the second use of the word 'be' so that it reads as '...factors might facilitate...". Additionally, please specify the ‘other pain-related variables’ that you mean.

Response: We have done so. We revised the Discussion section. We added the following text:

Line 427: Other pain-related factors (pain, fear, depression, self-efficacy, and body perception) should also be considered, as these factors might facilitate a reduction of the FRR or contribute to CLBP disability.

-Page 29, line 427: Please suggest some examples of interventions that might provide a reduction in the variability of muscle activity in people with CLBP.

Response: We have done so. We revised the Discussion section. We added the following text:

Line 447: No studies have been reported to date to examine whether the intervention can improve muscle activity variability reduction. However, EMG feedback therapy with high-density EMG is a potential future intervention [54]. Based on the association between pain and muscle activity, a decrease in pain and lumbar muscle stiffness has been reported after spinal manipulation [55]. Interventions that focus on muscle activity or on factors affecting muscle activity may be effective in improving muscle activity distribution variability.

-Page 30, line 443: I am not convinced that these interventions, or the findings from your study, justify this recommended. Please remove the words 'thus resolve' and replace with 'help reduce'

Response: We have done so. We revised the Discussion section. We added the following text:

Line 465: Physical therapy such as motor control exercises [57], spinal manipulation [58], and biofeedback training [59] to improve a low FRR could help reduce CLBP disability. 

-Page 31, line 456: Please replace the word 'should' with 'may'

Response: We have done so. We revised the Discussion section. We added the following text:

Line 478: Such comprehensive interventions may lead to much greater improvements of CLBP disability. A Patient whose LBP duration over 3 months. 

-Limitations: the sample size, multiple statistical analyses, and sub-grouping of participants are a concern and should be addressed as per the major feedback points above. Additionally, please consider adding the following as limitations: generalisability of results (based on methods e.g. FRR and participant characteristics).

Response: We have done so. We added further limitations. We revised the Abstract section. We added the following text:

Line 482: (1) The sample size of this study is not large. A small sample size increases the probability that one patient's results will be affected by small sample size in association rule analysis (epecially “support”). In the present study, the Fisher exact test was performed, one method of examining generalizability when the sample size is small. However, the current study results did not exclude the possibility that the sample size affected the results. (2) There are several problems with the generalizability of the results of this study. First, the small sample size. Second. the characteristics of this patient (diagnosis of LBP, area of LBP) may have influenced the results. Third, the results of FRR and muscle activity variability may only be based on the present analysis method.

Line 502: (7) The current study was a cross-sectional study, and it was not possible to design an intervention based on a longitudinal course. Developing an intervention based on a longitudinal time is necessary for future studies.

-Page 32, line 478: Please explain how the results of the study improve the disability of with CLBP? I don't believe any interventions were tested. Please consider rephrasing to relate to understanding of function/relationship between variables measured, or something similar.

Response: As you mentioned in your comment, no intervention was considered in this study. We rephrased that our findings could help clinical design interventions, which may improve the disability of numerous individuals with CLBP. We revised the Discussion section. We added the following text:

Line 512: Our findings could help clinical design interventions, which may improve the disability of numerous individuals with CLBP.

-Page 32, line 479: Please abbreviate chronic low back pain, as per previous sections of the manuscript.

Response: We have done so. We revised the Abstract section. We added the following text:

Line 512: Our findings could help clinical design interventions, which may improve the disability of numerous individuals with CLBP.

 

Manuscript ID PONE-D-20-19882, " Combined abnormal muscle activity and pain-related factors affect disability in patients with chronic low back pain: An association rule analysis," by Shigetoh et al. ' 

Responses to the comments of Reviewer #2

Reviewer #2: 

In general, the article is well versed, efficiently organized, translated the knowledge in elaborative manner and highlighted the dark zones in the muscle activity and pain-related areas using sEMG and open new horizons for researchers in this field. Although, this study is methodologically sound. However, the progression on previous work is not substantial, and some specific concerns are listed below.

Response: Thank you for reviewing our manuscript. Your comments were highly insightful and enabled us to greatly improve the quality of our manuscript. In the following pages are our point-by-point responses to each of the comments of the reviewers as well as your own comments. Revisions in the text are shown using red-colored font.

1. The introduction (rationale) need to be strengthen to highlight the significance of the present investigation. Like, there was little information about surface EMG, while this is the key part of the manuscript.

Response: Based on previous studies using electromyography (EMG), we have described the second paragraph was mainly about the flexion-relaxation ratio, and the third paragraph was about muscle activity distribution variability and the association between muscle activity and pain-related factors. We added the key word “EMG” and information on the EMG itself. Also, we added the relationship between decreased motor variability and tight control. We revised the Introduction section. We added the following text:

Line 64: Studies of CLBP patients reported that the patients' lumbar flexion-relaxation (FR) and variability of muscle activity distribution, which were observed by electromyography (EMG), were distorted [14,15,16].

Line 72: The variability of muscle activity distribution in CLBP patients is assessed by multi-channel EMG because the limited information that can be obtained from a single pair of electrodes placed over a small muscle region [18].

Line 75: These studies also indicated that the regions of muscle activation in CLBP patients were restricted and cranial regions were activated than caudal regions. This decreased motor variability is associated with tight motor control [21]. Increased trunk muscle activation to tighten control comes at the cost of increased spinal loading [22].

2. Page 11, Line 79-80: Can you split the sentences, “Several research groups have…”. Write each group’s individual work(s).

Response: We have done so. We revised the Introduction section. We added the following text:

Line 83: Several research groups have also described relationships between the variability of muscle activity distribution and physical factors in CLBP, including fatigue [19,20,29]. Focusing on the pain-related factors, one research group has also described relationships between the variability of muscle activity distribution and pain-related factors in CLBP, including pain severity [30] and disability [30].

3. Page 12, Line: 97: “Patients”: include demographic characteristics of men and women separately (age, weight, height etc.). And instead of male and female, change to men and women through the MS.

Response: We have done so. We revised the Methods section and Table 2. We added the following text:

Line 102: Twenty-four patients with CLBP (10 men, 14 women) aged 49–85 years (71.6 ± 9.2 yrs, mean ± standard deviation [SD], men: 71.5 ± 6.6, women: 71.7 ± 10.7) were referred from physiotherapy practices and enrolled if they were suffering from LBP lasting >3 months [32]. 

4. Page 16, section “Trunk kinematic record and analysis”: you did not mention about the any axis of the accelerometer (3-axis). How did you calculated the kinematic parameters? Has this device been validated before against lab standards? Change the figure 2 and put a real subject’s picture with accelerometer, also indicate the EMG grid and accelerometer. Line 162: Please change the sentence as “An accelerometer was fixed over the patient's twelfth thoracic (T12)…”

Response: In this study, the angle of trunk flexion was calculated using the angle between the gravity vector (g) and the z-axis vector (in this study, the direction along the trunk). The accelerometer used in this study (AX-3) has been validated in a previous study (Reference No.43). We added the picture in Figure 2. We revised the Methods section and Figure 2. We added the following text:

Line 172: This accelerometer has been validated in a previous study [43]. An accelerometer was fixed over the patient's twelfth thoracic (T12) spinous process with adhesive tape (Fig. 2).

Line 175: In this study, the angle of trunk flexion was calculated using the angle between the gravity vector and the z-axis vector (in this study, the direction along the trunk) [44].

5. The EMG was recorded with 1000 Hz sampling freq, and accelerometer with 100 Hz. So, how do you combine both the signals? Or, you have extracted/analyzed separately?

Response: We have extracted/analyzed the EMG and accelerometer separately. 

6. Is only Normalized RMS from EMG adequate to get the successful results?

Response: The present study focused on the flexion-relaxation ratio (FRR) and muscle activity distribution variability. For FRR, other muscle activity indices besides the normalized RMS were needed, such as maximum muscle activity to calculate it according to previous studies.

7. Page 20, line 244:: “Each binarized variable was classified into two groups, a 'good' group and a 'poor' group”, how do you scaled poor and good? Did you find any previous research work(s)?

Response: Previous studies have used the equal frequency discretization method and cluster analysis to classify them into two or three groups according to the magnitude of the score for each variable (Examples of descriptions in previous studies: http://usir.salford.ac.uk/id/eprint/53163). In describing the rules extracted in this study, we have also used the expressions "high" and "low." And previous studies have also interpreted this as good/poor, with higher scores indicating poorer conditions depending on each variable. The variables used in this study were classified as good and poor because labeling groups by high and low values does not allow for consistent interpretation, such as higher values for poorer values (e.g., pain intensity) and lower values for poorer values (e.g., PSEQ), depending on the variable. In the manuscript, we have also noted the variables with different interpretations of high and low values of the variables.(Line 262: Low values of the FRR, variability, and the PSEQ, and high values of other variables were classified as the poor group.)

---

## [Decision Letter · Decision Letter 1]

1 Dec 2020

PONE-D-20-19882R1

Combined abnormal muscle activity and pain-related factors affect disability in patients with chronic low back pain: An association rule analysis

PLOS ONE

Dear Dr. Shigetoh,

Thank you for submitting your manuscript to PLOS ONE. After careful consideration, we feel that it has merit but does not fully meet PLOS ONE’s publication criteria as it currently stands. Therefore, we invite you to submit a revised version of the manuscript that addresses the points raised during the review process.

One of the reviewers has suggested some minor corrections to the revised manuscript. Please address the comments in your response

We look forward to receiving your revised manuscript.

Kind regards,

Chris Connaboy

Academic Editor

PLOS ONE

Reviewers' comments:

Reviewer's Responses to Questions

**Comments to the Author**

1. If the authors have adequately addressed your comments raised in a previous round of review and you feel that this manuscript is now acceptable for publication, you may indicate that here to bypass the “Comments to the Author” section, enter your conflict of interest statement in the “Confidential to Editor” section, and submit your "Accept" recommendation.

Reviewer #1: (No Response)

Reviewer #2: All comments have been addressed

2. Is the manuscript technically sound, and do the data support the conclusions?

Reviewer #1: Yes

Reviewer #2: Yes

3. Has the statistical analysis been performed appropriately and rigorously? 

Reviewer #1: Yes

Reviewer #2: Yes

4. Have the authors made all data underlying the findings in their manuscript fully available?

Reviewer #1: Yes

Reviewer #2: Yes

5. Is the manuscript presented in an intelligible fashion and written in standard English?

Reviewer #1: Yes

Reviewer #2: No

6. Review Comments to the Author

Reviewer #1: Thank you for addressing my comments and providing a clear outline of how this was done. I have a few additional minor comments below:

Page 7, lines 113 to 115: The changes provide more detail about the study participants which will help inform readers in terms of applicability of findings to their patients, and researchers in terms of study replication or inclusion in systematic reviews. I have made some edits to these sentences and written it out as follows.

Also, the patient's LBP was categorized as non-specific LBP (n = 6) and specific LBP (n = 18). The specific LBP included spinal stenosis (n = 3) and lumbar osteoarthritis (n =15).

Page 12, line 197: please edit sentence as follows.

The patient's glabrous skin over the erector spinae was abraded and cleaned with an alcohol wipe.

Page 33, lines 490 to 498: I have made edits to these sentences, please see below.

This study had several limitations. (1) The sample size of this study is not large. A small sample size increases the probability that one patient's results will be affected by small sample size in association rule analysis (especially “support”). In the present study, the Fisher exact test was performed, one method of examining generalizability when the sample size is

small. However, the current study results did not exclude the possibility that the sample size affected the results. (2) The small sample size, the type of LBP experienced by participants, and the specific assessment techniques used limit the generalizability of this study's findings.

Reviewer #2: The authors carefully revised the manuscript and addressed all issues raised with respect to the previous version in a satisfactory way. The updated manuscript is now satisfactory in all aspects

7. PLOS authors have the option to publish the peer review history of their article (what does this mean?). If published, this will include your full peer review and any attached files.

Reviewer #1: No

Reviewer #2: No

---

## [Author Response · Author response to Decision Letter 1]

1 Dec 2020

2-December-2020

Dr. Joerg Heber

Editor-in-Chief

PLOS ONE

Dear Editors:

Manuscript ID PONE-D-20-19882, " Combined abnormal muscle activity and pain-related factors affect disability in patients with chronic low back pain: An association rule analysis," by Shigetoh et al. ' 

Please find attached a revised version of our above-named manuscript, which we would like to resubmit for publication as a Research Article in PLOS ONE.

Your comments and those of the reviewers were highly insightful and enabled us to greatly improve the quality of our manuscript. In the following pages are our point-by-point responses to each of the comments of the reviewers as well as your own comments.

Revisions in the text are shown using red-colored font. In accord with Reviewer No.1’s suggestion, we revised text in the Methods and Discussion. We hope that the revisions in the manuscript and our accompanying responses will be sufficient to make our manuscript suitable for publication in PLOS ONE.

We look forward to hearing from you at your earliest convenience.

Sincerely,

Dr. Hayato Shigetoh

Department of Neurorehabilitation

Graduate School of Health Sciences, Kio University

4 Chome-2-2 Umaminaka, Koryo, Kitakatsuragi District, Nara 635-0832, Japan

Tel.: +81-745-54-1601

Fax: +81-745-54-1600

Email: hayato.pt1121@gmail.com

Manuscript ID PONE-D-20-19882, " Combined abnormal muscle activity and pain-related factors affect disability in patients with chronic low back pain: An association rule analysis," by Shigetoh et al. ' 

Responses to the comments of Reviewer #1

Reviewer #1: 

Thank you for addressing my comments and providing a clear outline of how this was done. I have a few additional minor comments below:

Response: Thank you for reviewing our manuscript. Your comments were highly insightful and enabled us to greatly improve the quality of our manuscript. In the following pages are our point-by-point responses to each of the comments of the reviewers as well as your own comments. Revisions in the text are shown using red-colored font.

Page 7, lines 113 to 115: The changes provide more detail about the study participants which will help inform readers in terms of applicability of findings to their patients, and researchers in terms of study replication or inclusion in systematic reviews. I have made some edits to these sentences and written it out as follows.

Also, the patient's LBP was categorized as non-specific LBP (n = 6) and specific LBP (n = 18). The specific LBP included spinal stenosis (n = 3) and lumbar osteoarthritis (n =15).

Response: We have done so. We revised the Methods section.

Page 12, line 197: please edit sentence as follows.

The patient's glabrous skin over the erector spinae was abraded and cleaned with an alcohol wipe.

Response: We have done so. We revised the Methods section.

Page 33, lines 490 to 498: I have made edits to these sentences, please see below.

This study had several limitations. (1) The sample size of this study is not large. A small sample size increases the probability that one patient's results will be affected by small sample size in association rule analysis (especially “support”). In the present study, the Fisher exact test was performed, one method of examining generalizability when the sample size is

small. However, the current study results did not exclude the possibility that the sample size affected the results. (2) The small sample size, the type of LBP experienced by participants, and the specific assessment techniques used limit the generalizability of this study's findings.

Response: We have done so. We revised the Discussion section.

---

## [Decision Letter · Decision Letter 2]

3 Dec 2020

Combined abnormal muscle activity and pain-related factors affect disability in patients with chronic low back pain: An association rule analysis

PONE-D-20-19882R2

Dear Dr. Shigetoh,

We’re pleased to inform you that your manuscript has been judged scientifically suitable for publication and will be formally accepted for publication once it meets all outstanding technical requirements.

Kind regards,

Chris Connaboy

Academic Editor

PLOS ONE

Additional Editor Comments (optional):

Reviewers' comments:

Reviewer's Responses to Questions

**Comments to the Author**

1. If the authors have adequately addressed your comments raised in a previous round of review and you feel that this manuscript is now acceptable for publication, you may indicate that here to bypass the “Comments to the Author” section, enter your conflict of interest statement in the “Confidential to Editor” section, and submit your "Accept" recommendation.

Reviewer #1: All comments have been addressed

2. Is the manuscript technically sound, and do the data support the conclusions?

Reviewer #1: Yes

3. Has the statistical analysis been performed appropriately and rigorously? 

Reviewer #1: Yes

4. Have the authors made all data underlying the findings in their manuscript fully available?

Reviewer #1: Yes

5. Is the manuscript presented in an intelligible fashion and written in standard English?

Reviewer #1: Yes

6. Review Comments to the Author

Reviewer #1: Thank you for addressing the additional feedback. I think the changes have improved the paper and I look forward to seeing it published.

7. PLOS authors have the option to publish the peer review history of their article (what does this mean?). If published, this will include your full peer review and any attached files.

Reviewer #1: No

---

## [Editor Report · Acceptance letter]

7 Dec 2020

PONE-D-20-19882R2 

Combined abnormal muscle activity and pain-related factors affect disability in patients with chronic low back pain: An association rule analysis 

Dear Dr. Shigetoh:

I'm pleased to inform you that your manuscript has been deemed suitable for publication in PLOS ONE. Congratulations! Your manuscript is now with our production department. 

Kind regards, 

on behalf of

Dr. Chris Connaboy 

Academic Editor

PLOS ONE